# Association of ABC Efflux Transporter Genetic Variants and Adverse Drug Reactions and Survival in Patients with Non-Small Lung Cancer

**DOI:** 10.3390/genes16040453

**Published:** 2025-04-15

**Authors:** Cecilia Souto Seguin, Giovana Fernanda Santos Fidelis, Carolina Dagli-Hernandez, Pedro Eduardo Nascimento Silva Vasconcelos, Mariana Vieira Morau, Yasmim Gabriele Matos, Maurício Wesley Perroud, Eder de Carvalho Pincinato, Patricia Moriel

**Affiliations:** 1School of Medical Sciences, Universidade Estadual de Campinas, Campinas 13083894, Brazil; ceciliaseguin@gmail.com (C.S.S.); giovanafsfidelis@gmail.com (G.F.S.F.); pedro.nsvasconcelos89@gmail.com (P.E.N.S.V.); marianavmorau@gmail.com (M.V.M.); yg.matos@gmail.com (Y.G.M.); mperroud@unicamp.br (M.W.P.J.); edercp@unicamp.br (E.d.C.P.); 2Faculty of Pharmaceutical Sciences, Universidade Estadual de Campinas, Campinas 13083970, Brazil; carolina.hernandez@usp.br; 3Department of Pharmacy, School of Pharmaceutical Sciences, Universidade de São Paulo, São Paulo 05508000, Brazil

**Keywords:** ABC transporter, lung cancer, genetic variants

## Abstract

Background/Objectives: Lung cancer has a high mortality rate worldwide, with non-small cell lung cancer (NSCLC) being the most prevalent. Carboplatin and paclitaxel are key treatments for NSCLC; however, adverse drug reactions (ADRs) pose significant challenges. This study examined the impact of genetic variations in *ABCB1* and *ABCC2* genes on the incidence of ADRs and survival in NSCLC patients treated with carboplatin and paclitaxel. Methods: Variants were identified using RT-PCR, and ADRs classified according to the Common Toxicity Criteria for Adverse Events, Version 4.03. Results: The *ABCB1* rs1128503 (c.1236C>T) CC genotype was associated with a higher chance of nausea (OR: 3.5, 95% CI 1.367–9.250, *p* = 0.0093), vomiting (OR: 13.553, 95% CI 1.705–107.723, *p* = 0.0137), and a higher risk of death in CT or TT genotypes (HR: 1.725, 95% CI 1.036–2.871, *p* = 0.0361). The *ABCC2* rs717620 (c.-24C>T) TT genotype was associated with increased ALP levels (OR: 14.6, 95% CI 1.234–174.236, *p* = 0.0335). The *ABCB1* rs2032582 non-CC genotypes (TT+AA+TA+CA+CT) were associated with an increased risk of death (HR: 1.922, 95% CI 1.093–3.377, *p* = 0.0232). Patients with hypocalcemia (HR: 2.317, 95% IC 1.353–3.967, *p* = 0.022), vomiting (HR: 3.047, 95% IC 1.548–5.997, *p* = 0.0013), and diarrhea (HR: 2.974, 95% IC 1.590–5.562, *p* = 0.0006) were associated with lower overall survival. Conclusions: The data suggest that *ABCB1* variants may influence gastrointestinal ADRs and patient survival, highlighting the importance of pharmacogenomics in predicting ADRs and drug resistance. This approach offers more precise pharmacotherapy, reduces ADRs, and enhances the patients’ quality of life and survival.

## 1. Introduction

Lung cancer is the leading cause of cancer-related morbidity and mortality worldwide. The American Cancer Society estimates that approximately 340 individuals die of lung cancer every day in the United States, with a projection of 124,730 deaths in 2025 [1]. In Brazil, a total of 28.964 deaths due to lung cancer was observed in 2021 [2]. Non-small cell lung carcinoma (NSCLC) accounts for 85% of lung cancer cases and has a poor prognosis, as evidenced by an estimated 5-year survival of 26.4% [3]. The main risk factor for NSCLC development is tobacco smoking, which contributes to approximately 90% of the cases [4].

According to the guidelines published by the European Society of Medical Oncology for NSCLC, the use of carboplatin continues to be considered as the gold standard in the treatment of NSCLC [5,6]. Although carboplatin-paclitaxel is a widely used therapeutic regimen, it can cause multiple adverse drug reactions (ADRs), resulting in dose reductions or even the temporary discontinuation of chemotherapy. This can ultimately lead to treatment delays and compromise both disease control and patient survival [7,8].

Genetic variations in influx and efflux transporters involved in the pharmacokinetics of carboplatin and paclitaxel may alter their efficacy and increase the risk of ADRs associated with chemotherapy [9,10]. ATP-binding cassette (ABC) transporters comprise a superfamily of proteins that use ATP hydrolysis energy to transport endogenous and exogenous substances across the cell membrane and are expressed in the kidneys, lungs, liver, and small intestine [11]. *ABCB1* and *ABCC2* genes encode two drug transporters, ABCB1 and ABCC2, involved in the transport of platinum [12]. Variants in these genes may influence the absorption, distribution, and excretion of these drugs, resulting in the intracellular accumulation of platinum and variability in paclitaxel pharmacokinetics, which may, in turn, contribute to the development of ADRs [13,14].

Recently, several studies have evaluated the influence of SNVs in ABC transporter genes on the development of ADRs in response to chemotherapy [15]. ABCB1 has been extensively described in the literature, with its most studied genetic variants being rs1045642 (c.3435A>G), rs1128503 (c.1236C>T), and rs2032582 (c.2677A>C/T) [16]. These variants have been associated with altered ABCB1 expression or activity in in vitro studies [17,18,19,20,21,22,23]. Genotyping of this transporter is highly relevant in oncological treatments [10,24,25,26,27]. For example, the *ABCB1* rs1128503 (c.1236C>T) genotype is strongly associated with hepatic ADRs in patients with lung cancer undergoing platinum-based doublet chemotherapy [11]. Another study reported that the heterozygous genotype AG of *ABCB1* rs1045642 (c.3435A>G) was associated with a lower risk of hematological toxicity compared to homozygous CC [28]. Concerning the *ABCC2* transporter, the *ABCC2* rs717620 (c.24C>T) variant has been demonstrated to be associated with overall survival (OS) in patients with advanced NSCLC [29].

Given the pharmacogenetic importance of *ABCB1* and *ABCC2* variants, this study aimed to identify the mechanisms by which genetic variations may contribute to the induction of ADRs and impact survival in patients with NSCLC treated with carboplatin and paclitaxel.

## 2. Materials and Methods

### 2.1. Study Design and Set-Up

This prospective cohort study was conducted at the Onco-pneumology Ambulatory Hospital of the Universidade Estadual de Campinas (Campinas, São Paulo, Brazil). The patients were recruited from March 2018 to October 2022. Patients diagnosed with NSCLC were followed-up to assess the occurrence of hematological, renal, hepatic, and gastrointestinal ADRs during the first 21-day cycle of chemotherapy with carboplatin and paclitaxel. Clinical and demographic data were collected via interviews and medical charts reviews. Hematological and biochemical tests were performed before (D0) and 21 days after (D20) the first chemotherapy session.

This study was approved by the Ethics Committee of the Universidade Estadual de Campinas (protocol code: 83196318.8.0000.5404) and written informed consent was obtained from each patient.

### 2.2. Patients

Patients aged ≥18 years with histologically confirmed NSCLC, a Karnofsky Performance Status (KPS) ≥ 60%, and who were scheduled to receive their first 21-day chemotherapy cycle with paclitaxel (200 mg/m^2^) in combination with carboplatin (AUC = 5/6 according to Calvert’s formula) were included. Only genetically unrelated individuals were included in the study. Patients were excluded if they had already received prior chemotherapy; received the first paclitaxel and carboplatin cycle at another hospital; were diagnosed with hepatitis, HIV, or a psychiatric disorder that would limit their communication; had undergone any change in the chemotherapy protocol during the treatment; or refused to participate in the study.

Sample size was calculated using the Raosoft^®^ sample size calculator http://www.raosoft.com/samplesize.html (accessed on 5 April 2025). For this analysis, we considered a response distribution of 19%, which is the frequency of *ABCC2* rs717620 (c.-24C>T), a power of 80%, and margin of error of 5%, resulting in a minimum sample size of 101 patients.

### 2.3. Data Collection

Demographic and clinical data were collected through interviews and medical chart reviews. Demographic data included age, sex, comorbidities, self-reported ethnicity, and continuous use of medications. Smoking status was assessed using the smoking index (SI), which was determined by multiplying the number of cigarettes smoked per day with the total number of years smoked [30] (Jindal et al., 1982). Patients were classified as non-smokers (SI = 0), light smokers (SI = 1–100), moderate smokers (SI = 101–200), and heavy smokers (SI ≥ 300). Alcohol consumption was classified according to the scale by Whitcomb et al. (2008) [31], which evaluates the number of alcohol doses consumed during the period of maximal consumption in a patient’s life [32]. Patients were classified as abstainers (<20 doses throughout their entire life), light drinkers (≤3 doses/week), moderate drinkers (women: 4–7 doses/week; men: 4–14 doses/week), heavy drinkers (women: 8–34 doses/week; men: 15–34 doses/week), and very heavy drinkers (≥35 doses/week).

Clinical data included body mass index and KPS, as determined by the medical team [33]. Tumor histological type and size, as well as the presence of metastasis, were determined by the medical team and collected from the medical charts. The tumor stage was classified using the TNM system.

### 2.4. Adverse Reactions Assessment

To determine the occurrence of ADRs, blood samples were collected before (D0) and after (D20) the first 21-day paclitaxel and carboplatin chemotherapy cycle. Hematological, renal, hepatic, and gastrointestinal ADRs, and their respective grades were determined using the Common Terminology Criteria for Adverse Events (CTCAE; v.4) (National Cancer Institute, 2009).

The following hematological ADRs were assessed: anemia, leukopenia, neutropenia, lymphopenia, and thrombocytopenia. For renal ADRs, the following events were recorded: hyperuricemia, hyponatremia, hypomagnesemia, hypokalemia, hypophosphatemia, hypocalcemia, increased serum creatinine levels, and reduced creatinine clearance. The evaluated hepatic parameters included hypoalbuminemia and increased aspartate aminotransferase (AST), alanine aminotransferase (ALT), alkaline phosphatase (ALP), and total bilirubin (TB). Data on gastrointestinal ADRs, nausea, vomiting, and diarrhea were collected through patient interviews and medical chart reviews.

### 2.5. ABCB1 and ABCC2 Genotyping

Genomic DNA was extracted from peripheral blood and purified using a Wizard^®^ Genomic DNA Purification Kit (Promega, Madison, WI, USA). DNA concentration and integrity were verified by analyzing the A_260/280_ ratio with a NanoDrop-1000-Detector (Thermo Fisher Scientific, Waltham, MA, USA). Definitive quantification was performed using the Quantus fluorometer^TM^ and the respective kit (QuantiFluor^TM^ dsDNA System, Promega, Madison, WI, USA). All samples were identified and adjusted for 20 ng/µL and were later stored at −20 °C for further use.

*ABCB1* rs1045642 (c.3435A>G), rs1128503 (c.1236C>T), rs2032582 (c.2677A>C/T), and *ABCC2* rs717620 (c.-24C>T) were genotyped using Taqman^®^ assay (Thermo Fisher Scientific, Waltham, MA, USA). The amplification reaction mixture (10 µL) contained 2 µL DNA (20 ng/µL), 5 µL TaqMan™ Genotyping Master Mix, 2.5 µL DNAse-RNAse free water, and 0.5 µL TaqMan™ SNP Genotyping assays. A real-time polymerase chain reaction (RT-PCR) was performed using the Rotor-Gene Q 5plex HRM System (Qiagen, Hilden, Germany) with an initial phase of 2 min at 60 °C to acclimatize the reagents, an enzymatic activation phase of 10 min at 95 °C, followed by 50 cycles of denaturation, annealing, and extension at 95 °C for 15 s and 60 °C for 1 min. Allelic discrimination was performed using Rotor-Gene Q series software (Version 2.3.0).

### 2.6. Statistical Analysis

Categorical variables were expressed as number and frequency. Continuous variables were expressed as mean ± standard deviation. Fisher’s exact test, univariate logistic regression, and Cox regressions were conducted to analyze the associations between clinical and demographic variables (age, sex, ethnicity, smoking, alcohol consumption, and presence of comorbidities), *ABCB1* and *ABCC2* variants, and ADRs. The Hardy–Weinberg equilibrium (HWE) was tested using the χ^2^ goodness-of-fit test (*p* ≥ 0.05). Significant associations (*p* < 0.05) were further tested in multivariate logistic analyses, using the “Stepwise regression” method, for the calculation of adjusted odds ratios (OR_adjusted_) and respective 95% confidence intervals (95% CI). Univariate and multivariate logistic regression analyses were performed using the recessive and dominant models. Relevant clinical variables, such as genetic variants, age, sex, ethnicity, smoking, alcoholism, and comorbidities, were considered covariates in the multivariate analysis. The dominant (AA vs. AG+GG; CC vs. CT+TT) and recessive (GG vs. AG+AA; TT vs. CT+CC) models were utilized to target *ABCB1* rs1045642 (c.3435A>G), *ABCB1* rs1128503 (c.1236C>T), and *ABCC2* rs717620 (c.-24C>T). For rs2032582 (c.2677A>C/T), the models were designed as CC vs. non-CC (TT+AS+TA+CA+CT), and AA vs. non-AA (TT+CC+AT+CA+CT). Individual features (clinical and histopathological variables and genotypes) were evaluated for their association with the occurrence of adverse reactions using the Chi-square test or Fisher’s exact test (categorical data). Survival and OS were estimated using a Kaplan–Meier analysis, and factors associated with survival were estimated using a multiple Cox regression analysis. OS was measured from the time of diagnosis until death or the last follow-up. The results were considered statistically significant at *p* < 0.05. Statistical tests were performed using SAS v. 9.4 (SAS Institute Inc., 2002–2012, Cary, NC, USA). 

## 3. Results

### 3.1. Demographic and Clinical Characteristics of Patients

One hundred and seventy-seven (177) NSCLC patients who qualified through the inclusion criteria for this study started their first chemotherapy cycle with the carboplatin and paclitaxel protocol. Out of 177, 65 patients were excluded because of several reasons, such as death before the initiation of protocol (15), a change in protocol (19), chemotherapy suspension (14), patient withdrawal (3), and incomplete data collection due to the COVID-19 pandemic (14), totaling 113 patients to be included for ADR analysis. Furthermore, blood samples from three patients were not available for DNA extraction, reducing the total number of patients who were eligible for *ABCB1* and *ABCC2* genotyping to 110.

The patients were predominantly males (53.1%), with a mean age of 63 years, whites (82%), light drinkers (34.5%), and heavy smokers (38.1%) (Table 1). Most patients had no comorbidities (57.5%) and had not undergone previous surgery (96.5%). Adenocarcinoma was found to be the most common histological type (58.4%). According to the TNM classification, the tumors were mostly categorized as T4 (69.9%), N2 (47.8%), and M1 (48.7%) (Table 2).

### 3.2. Clinical Parameters

The clinical parameters for the assessment of hematological, renal, and hepatic ADRs are depicted in Appendix A. Statistically significant reductions were observed between baseline (D0) and D20 for the hematological parameters, including hemoglobin, leukocytes, neutrophils, and platelets. Reductions were also observed in the renal parameters of calcium, magnesium, and urea, as well as in the hepatic parameters of TB and total proteins.

### 3.3. Occurrence and Severity of ADRs

In this cohort, 93 (82.3%) patients experienced ADRs. Of these, 60 (53.1%) had renal ADRs, 59 (52.2%) exhibited hematological reactions, 39 (34.5%) suffered from gastrointestinal events, and 36 (31.9%) showed symptoms of hepatic ADRs.

The observed ADRs and their grades, according to the CTCAE classification (CTCAE; v.4), are described in Table 3. The most frequent ADRs were anemia (39.8%), nausea (26.5%), hypocalcemia (26.3%), and reduced creatinine clearance (25.2%), whereas the most severe ADR was grade 4 leukopenia, which affected one (0.9%) patient. However, most events were classified as grade 1 or 2.

### 3.4. Frequency of ABCB1 and ABCC2 Variants

The frequencies of *ABCB1* rs1045642 (c.3435A>G), rs1128503 (c.1236C>T), rs2032582 (c.2677A>C/T), and *ABCC2* rs717620 (c.-24C>T) genotypes and alleles are shown in Appendix A. The distributions of all genotypes were in HWE (*p* ≥ 0.05).

Regarding *ABCB1* rs1128503 (c.1236C>T), 45.5% of the patients were homozygous for the variant C allele (CC), whereas for *ABCB1* rs1045642 (c.3435A>G), the majority were heterozygous for AG (46.2%). Most patients were heterozygous carriers of the AC, CT, or AT genotypes (45.4%) of the *ABCB1* triallelic variant, rs2032582 (c.2677A>C/T). C allele was the most frequent (65.7%), with 44.4% homozygous carriers (CC) (Appendix A).

For the *ABCC2* rs717620 c.-24C>T variant, most patients were homozygous for the reference allele C (CC, 70.0%). The allele frequency observed in this study closely corresponded to that reported for the Brazilian population. Statistically significant differences (*p* < 0.0001) were found between the global and Brazilian allele frequency of *ABCB1* gene variants (Appendix A).

### 3.5. Association of Clinical and Demographic Data with ADRs

A univariate logistic regression was performed to analyze the associations between clinical characteristics and ADR development. For this analysis, only CTCAE grade ≥1 ADRs that occurred in more than 10% of the patients were included. The following clinical and demographic variables were analyzed: age, sex, ethnicity, smoking status, alcohol consumption, and presence of comorbidities.

All univariate logistic regression analyses are reported in Appendix A. Women were more likely than men to experience nausea (OR: 2.489, 95% CI 1.051–5.894, *p* = 0.0381). Advanced age increased the frequency of hepatic ADR hypoalbuminemia (OR: 1.091, 95% CI 1.011–1.178, *p* = 0.0250) and renal ADR hypocalcemia (OR: 1.102, 95% CI 1.025–1.185, *p* = 0.0087). Non-smokers also had more hepatic reactions, observed by ALP increase (OR: 4.6, 95% CI 1.409–15.258, *p* = 0.0116). No association was found between the remaining ADRs and demographic and clinical variables. These significant associations (*p* < 0.05) were analyzed using a multivariate regression analysis; however, no statistically significant associations were observed.

### 3.6. Association of ABCB1 and ABCC2 Variants with ADRs

Univariate logistic regression analyses were performed to verify the association between ADRs and *ABCB1* and *ABCC2* variants (Appendix A).

The *ABCC2* rs717620 (c.-24C>T) genotype demonstrated a significant association with hepatic ADRs in a univariate logistic regression analysis (Appendix A). In the recessive model (TT vs. CT+CC), patients carrying the TT genotype were more likely to show an increase in ALP (OR: 14.3, 95% CI: 1.211–169.086, *p* = 0.0347), compared to CT and CC genotype carriers. This association remained significant after a multivariate analysis (OR: 14.6, 95% CI 1.234–174.236; *p* = 0.0335) (Table 4).

Variants in *ABCB1* were associated with gastrointestinal ADRs (Appendix A). Individuals carrying *ABCB1* rs1128503 (c.1236C>T) CC+CT genotypes had a higher risk of developing nausea (OR: 3.5, 95% CI 1.367–9.250, *p* = 0.0093) and vomiting (OR: 13.553, 95% CI 1.705–107.723, *p* = 0.0137) than those with the TT genotype. Additionally, CT+TT genotypes were associated with an increased risk of experiencing diarrhea (*p* = 0.038); however, none of the individuals with CC reported having diarrhea. Similarly, *ABCB1* rs2032582 (c.2677A>C/T) non-AA genotypes (TT+CT+AT+CA+CT) were associated with nausea (*p* = 0.0341); however, none of the individuals with AA reported nausea. Since no cases were found for either of the aforementioned variants, the odds ratios could not be estimated. Despite these notable associations identified in the univariate logistic regression analyses, no significant association was found in the multivariate analysis.

Other ADRs, such as hematological and renal ADRs, were not associated with *ABCB1* and *ABCC2* variants in this analysis (Appendix A).

### 3.7. Survival Analysis

A total of 65 (57.5%) patients had died by the end of the study period. The average OS was 1.68 ± 0.15 years, equivalent to approximately 1 year and 8 months. The probability of survival was 2.58 years (approximately 2 years and 7 months) for 75%, 1.21 years (1 year and 2 months) for 50%, and 0.64 years (8 months) for 25% of the patients (Appendix A).

OS was assessed using a univariate Cox regression analysis, examining the association between clinical and demographic data or ADRs and survival. Only ADRs that occurred in more than 10% of patients with grade ≥1 were included in the analysis.

The results of the univariate Cox regressions associating clinical and demographic data with survival are elucidated in Appendix A. The analysis showed that smokers had a 5.048 times greater risk of death than non-smokers (95% CI 2.004–12.711, *p* = 0.0006). The other clinical and demographic parameters were not associated with survival.

The risk of death associated with ADRs is presented in Appendix A. Patients with hypocalcemia showed a 2.317 times higher risk of death than those without hypocalcemia (95% CI 1.353–3.967, *p* = 0.022). Additionally, patients who experienced vomiting and diarrhea showed 3.047 (95% CI 1.548–5.997, *p* = 0.013) and 2.974 (95% CI 1.590–5.562, *p* = 0.0006) times greater risks of death, respectively. The other ADRs were not associated with survival (Appendix A).

The risk of death associated with *ABCB1* and *ABCC2* variants is described in Appendix A. Death associated with *ABCB1* variants is shown in Figure 1 and Figure 2. Patients with *ABCB1* rs2032582 non-CC (TT+AA+TA+CA+CT) genotypes had a higher risk of death when compared with patients carrying the CC genotype (HR: 1.795, 95% CI: 0.836–3.784, *p* = 0.0265). The multivariate analysis, which adjusted for genetic variants, age, sex, ethnicity, smoking, alcoholism, and comorbidities, still revealed an increased risk of death (HR: 1.922, 95% CI 1.093–3.377, *p* = 0.0232) (Table 5). Likewise, patients carrying *ABCB1* rs1128503 CT or CC genotypes had a higher risk of death than those carrying the TT genotype (HR: 1.725, 95% CI 1.036–2.871, *p* = 0.0361). However, no association was found in the multivariate regression analysis. Other genotypes and variants showed no significant associations with survival (Appendix A).

## 4. Discussion

The carboplatin–paclitaxel regimen remains a cornerstone of chemotherapy in adjuvant, neoadjuvant, and palliative settings for NSCLC. Since its introduction over two decades ago, it has been extensively documented in clinical trials and case reports [34]. Despite significant advances in new therapies for NSCLC, platinum-based chemotherapy remains the standard treatment, recommended in international guidelines, such as the European Society for Medical Oncology for advanced/metastatic NSCLC guideline [6]. However, the success of this therapy may be compromised by the development of chemotherapy-induced ADRs [35].

In our study, 52.2% of ADRs were of hematological origin, with anemia being the most frequent, and grade 4 leukopenia being the most severe. Hematological toxicity was the most frequent ADR in other clinical studies where patients received carboplatin–paclitaxel treatment, with the all-grade anemia frequency ranging from 50 to 100% [33,36,37,38]. However, no association was identified between hematological toxicity and *ABCB1* or *ABCC2* variants, aligning with the findings of an Indian cohort [39].

Chemotherapy can also induce nausea and vomiting, which affect approximately 40% of patients [40], mostly women [41]. In our study, nausea was the second most frequent ADR (26.5%) and was more prevalent in women, corroborating previous studies [42,43]. Gastrointestinal events are caused by oxidative stress generated by platinum-based compounds, which damage the enterochromaffin cells of the small intestine [44]. This leads to the release of emetic neurotransmitters, including serotonin, dopamine, and prostaglandins, that can directly affect specific receptors in the enteric nervous system and intestinal smooth muscle. Alternatively, these neurotransmitters can indirectly affect these receptors by activating emetic nuclei in the central nervous system [45].

In our study, patients with the *ABCB1* rs1128503 (c.1236C>T) CC+CT genotypes had a higher risk of developing nausea and vomiting. A previous study reported different results, as the CT genotype was associated with a reduced risk of nausea or vomiting compared to the TT genotype [39]. Furthermore, *ABCB1* rs1128503 CT and CC genotypes were associated with lower OS in our cohort. Similarly, other studies found that the TT genotype, alone or in a haplotype, was associated with longer OS [46,47,48,49,50]. Additionally, variants in the *ABCB1* gene have been associated with the response to taxane-based treatment. In particular, the *ABCB1* rs2032582 and *ABCB1* rs1045642 variants have been correlated with a partial response to these agents, suggesting a potential functional impact of these variants on therapeutic efficacy [10].

Notably, while *ABCB1* rs1128503 was associated with nausea, vomiting, and a lower OS, vomiting itself was also associated with lower OS in our cohort. These results suggest a possible chain of events: *ABCB1* rs1128203 may increase the risk of vomiting with carboplatin and paclitaxel treatment, which in turn leads to a lower OS in patients with NSCLC. Thus, we propose a mechanism for the induction of nausea and vomiting caused by the interplay between carboplatin, paclitaxel, and the *ABCB1* rs1128503 variant. Given that previous studies have shown that rs1128503 T allele might decrease ABCB1 activity [51,52], normal ABCB1 activity, due to the presence of the C (reference) allele, may result in higher efflux, and, as a result, higher plasma levels of carboplatin and paclitaxel, activating emetic pathways. Emetic neurotransmitters that trigger nausea and vomiting can compromise the effectiveness of the treatment, as high degrees of nausea and vomiting can lead to dose reduction and/or the temporary suspension of treatment. This may lead to disease progression, and consequently, a reduction in survival. Additionally, the development of nausea and vomiting affects electrolyte balance, causing dehydration and malnutrition, and significantly impacting the physical and psychological well-being of the patients [51]. Consequently, this event potentially has a substantial impact on the quality of life and effectiveness of the treatment [52].

For the *ABCB1* rs2032582 (c.2677A>C/T) variant, our study revealed an association between non-AA genotypes (namely, the AT/CC/CA/CT genotypes) and worse OS, which was further confirmed in the multivariate analysis. Likewise, the c.2677AA genotype was associated with a longer OS in Belgian patients with ovarian cancer treated with paclitaxel and carboplatin [53]. In Korean and Indian cohorts, CC/CT/TT genotypes were associated with a lower progression-free survival (PFS) compared to AA/AC/AT genotypes, but not with OS [27,36]. Therefore, the contribution of *ABCB1* rs2032582 to the risk of chemotherapy-induced ADRs needs to be further investigated.

*ABCC2* encodes the *ABCC2* transporter or multi-drug resistance protein 2 (MRP2), which is a well-established transporter of platinum and taxanes. Genetic variations in *ABCC2* activity may affect plasma drug levels and clearance, ultimately affecting drug response [27,54]. Patients carrying the TT genotype of *ABCC2* rs717620 had a 14-fold higher risk of developing hepatic ADR associated with elevated ALP levels. To the best of our knowledge, this is the first study to associate this variant with the risk of hepatic ADRs. Further studies are required to confirm these findings.

This study has two significant limitations. First, owing to challenges in maintaining patient follow-up during the COVID-19 pandemic, only the first chemotherapy cycle was evaluated. However, all patients included in this study were diagnosed by the same medical team, which minimized the potential for bias in the study population. The second limitation concerns the analysis of genetic variants. Only the impact of these variants on the development of ADRs was considered, without accounting for additional genes that could influence the outcomes. Addressing this would require a different type of analysis such as a genome-wide association study, which would require a substantially larger patient cohort to achieve the desired level of statistical significance [55]. Nonetheless, despite not accounting for these variables, this study identified a clear association between the investigated variants and ADRs, which was further confirmed through the multivariate analysis.

## 5. Conclusions

The findings of our study indicate that variants in the *ABCB1* gene may be associated with nausea and vomiting. Consequently, patients who experienced episodes of vomiting exhibited reduced survival rates. This finding suggests the importance of antiemetic therapy before and after chemotherapy sessions for preventing gastrointestinal ADRs.

Furthermore, this study demonstrates the potential of *ABCB1* variants as pharmacogenetic markers of ADRs in patients with NSCLC treated with carboplatin and paclitaxel. Pharmacogenetic testing before chemotherapy may reduce the risk of ADRs, potentially increasing treatment adherence and efficiency, and ultimately leading to better OS in these patients.

To the best of our knowledge, this is the first study to investigate the association between ABC transporter genetic variants and chemotherapy outcomes in lung cancer patients treated with carboplatin and paclitaxel in the Brazilian population. Given the highly heterogenous nature of the Brazilian population and its underrepresentation in pharmacogenetic studies [56,57,58], research in this group is essential for advancing pharmacogenetics implementation in countries with admixed populations. Further studies should explore the impact of genetic variants in genes involved in the pharmacokinetics and pharmacodynamics of carboplatin and paclitaxel to better elucidate their role in ADRs and OS, particularly in the Brazilian population. Additionally, larger sample sizes are needed to validate these findings in future research. 

## Figures and Tables

**Figure 1 genes-16-00453-f001:**
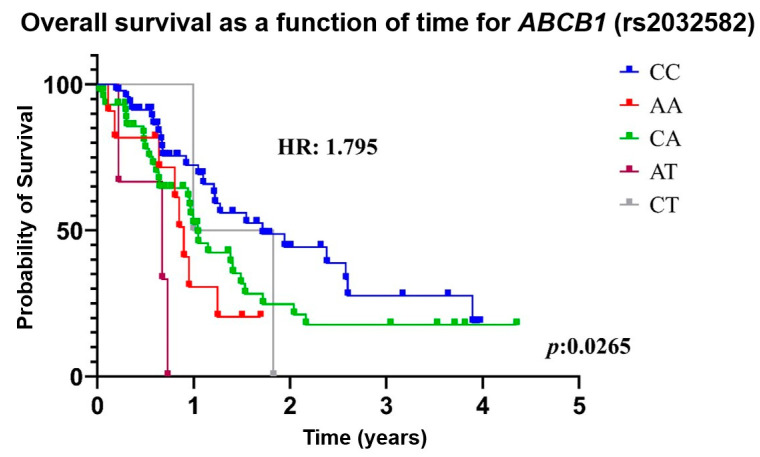
Kaplan–Meier curves illustrating the association between overall survival in different genotypes of *ABCB1* rs2032582 in patients with non-small cell lung cancer who were treated with carboplatin and paclitaxel. HR, hazard ratio; *ABCB1*, ATP Binding Cassette Subfamily B Member 1.

**Figure 2 genes-16-00453-f002:**
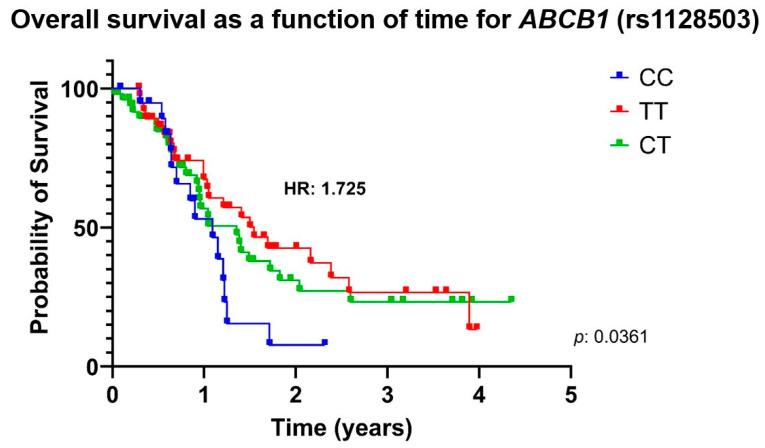
Kaplan–Meier curves illustrating the association between overall survival in different genotypes of *ABCB1* rs1128503 in patients with non-small cell lung cancer who were treated with carboplatin and paclitaxel. HR, hazard ratio; *ABCB1*, ATP Binding Cassette Subfamily B Member 1.

**Table 1 genes-16-00453-t001:** Demographic data of patients with non-small cell lung cancer treated with carboplatin and paclitaxel included in the study (*n* = 113).

Variable	Patients *n* (%)	
Sex		
Male	60	(53.1)
Female	53	(46.9)
Age (mean ± SD, years)	63.23 ± 7.31	
Ethnicity		
White	93	(82.3)
Non-white	20	(17.7)
Smoking status		
Non-smoker	17	(15.0)
Light smoker	38	(33.6)
Moderate smoker	15	(13.3)
Heavy smoker	43	(38.1)
Alcoholism status		
Abstainer	38	(33.6)
Light drinker	39	(34.5)
Moderate drinker	12	(10.6)
Heavy drinker	16	(14.2)
Very heavy drinker	8	(7.1)
Continuous-use medications		
Losartan	24	(21.2)
Formoterol + Budesonide	19	(16.8)
Simvastatin	16	(14.2)
Morphine	15	(13.3)
Metformin	14	(12.4)
Dipyrone	12	(10.6)
Hydrochlorothiazide	10	(8.8)
Acetylsalicylic acid	9	(8.0)
Dexamethasone	9	(8.0)
Amitriptyline	8	(7.1)

*n*, absolute number of patients; SD, standard deviation.

**Table 2 genes-16-00453-t002:** Clinical data of patients with non-small lung cancer who were treated with carboplatin and paclitaxel included in the study (*n* = 113).

Variable	Patients *n* (%)	
BMI (mean ± SD, kg/m^2^) Overweight (25.0–29.9)	24.40 ± 5.61	
	45	(39.8)
Normal (18.5–24.9)	62	(54.9)
Underweight (<18.5)	6	(5.3)
Performance status		
KPS 100%	101	(89.4)
KPS 90%	8	(7.1)
KPS 80%	2	(1.8)
KPS 60%	2	(1.8)
Comorbidities		
No comorbidities	65	(57.5)
Hypertension	36	(31.9)
Diabetes	21	(18.6)
Hypercholesterolemia	15	(13.3)
Other	5	(4.4)
Surgical resection before treatment		
No	109	(96.5)
Yes	4	(3.5)
Histological type		
Oat cell (small cell carcinoma)	1	(0.9)
SCC (squamous cell carcinoma)	38	(33.6)
Adenocarcinoma	66	(58.4)
Neuroendocrine	1	(0.9)
Undifferentiated	7	(6.2)
Primary tumor size—T		
T1	3	(2.7)
T2	9	(8.0)
T3	17	(15.0)
T4	79	(69.9)
Unidentified	5	(4.4)
Lymph node metastasis—N		
N0	11	(9.7)
N1	4	(3.5)
N2	54	(47.8)
N3	33	(29.2)
Unidentified	11	(9.7)
Presence of metastasis—M		
M0	34	(30.1)
M1	55	(48.7)
MX	24	(21.2)

*n*, absolute number of patients; SD, standard deviation; KPS, Karnofsky Performance Status; BMI, body mass index.

**Table 3 genes-16-00453-t003:** Frequency and severity of adverse drug reactions in patients with non-small lung cancer who were treated with carboplatin and paclitaxel according to Common Terminology Criteria for Adverse Events (CTCAE; v.4) grade.

ADRs	Total of Patients (*n*)	Total of Patients Who Experienced ADRs, *n* (%)	Grade 0 *n* (%)	Grade 1 *n* (%)	Grade 2 *n* (%)	Grade 3 *n* (%)	Grade 4 *n* (%)
Hematological							
Anemia	113	45 (39.8)	68 (60.2)	43 (38.0)	2 (1.8)	-	-
Leukopenia	113	20 (17.7)	93 (82.3)	11 (9.7)	6 (5.3)	2 (1.8)	1 (0.9)
Neutropenia	111	6 (5.4)	105 (94.6)	5 (4.5)	-	1 (0.9)	-
Lymphopenia	113	4 (3.6)	109 (96.4)	2 (1.8)	2 (1.8)	-	-
Thrombocytopenia	112	13 (11.6)	99 (88.4)	13 (11.6)	-	-	-
Renal							
Hyperuricemia	105	6 (5.7)	99 (94.3)	6 (5.7)	-	-	-
Hyponatremia	112	23 (20.5)	89 (79.5)	23 (20.5)	-	-	-
Hypomagnesemia	86	14 (16.3)	72 (83.7)	13 (15.1)	1 (1.2)	-	-
Hypokalemia	113	4 (3.5)	109 (86.5)	3 (2.7)	-	1 (0.9)	-
Hypophosphatemia	88	-	88 (100)	-	-	-	-
Hypocalcemia	99	26 (26.3)	73 (73.7)	26 (26.3)	-	-	-
Increased serum creatinine	112	5 (4.5)	107 (95.5)	5 (4.5)	-	-	-
Reduced creatinine clearance	111	28 (25.2)	83 (74.8)	27 (24.3)	1 (0.9)	-	-
Hepatic							
Hypoalbuminemia	113	19 (16.8)	94 (83.2)	18 (15.9)	1 (0.9)	-	-
AST increase	113	4 (3.5)	109 (96.5)	3 (2.6)	1 (0.9)	-	-
ALT increase	113	6 (5.3)	107 (94.7)	4 (3.5)	-	2 (1.8)	-
ALP increase	113	16 (14.1)	97 (85.8)	14 (12.4)	2 (1.8)	-	-
TB increase	113	-	113 (100)	-	-	-	-
Gastrointestinal							
Nausea	113	30 (26.5)	83 (73.4)	27 (23.9)	2 (1.8)	1 (0.9)	- *
Vomiting	113	16 (14.1)	97 (85.8)	13 (11.5)	2 (1.8)	1 (0.9)	-
Diarrhea	113	17 (15.0)	96 (84.9)	16 (14.2)	1 (0.9)	-	-

*n*, absolute number of patients; ADR, adverse drug reaction; ALT, alanine aminotransferase; ALP, alkaline phosphatase; AST, aspartate aminotransferase; TB, total bilirubin. * There is no grade 4 nausea according to CTCAE (version 4).

**Table 4 genes-16-00453-t004:** Multivariate logistic regression of adverse drug reactions in patients with non-small lung cancer who were treated with carboplatin and paclitaxel and *ABCB1* and *ABCC2* variants.

Variable	ADRs		*p*-Value *	OR (95% CI)
	Grade 0 (*n*, %)	Grade 1–4 (*n*, %)		
Increased ALP (*n* = 90)				
*ABCC2 rs717620 (c.-24C>T)*				
TT (ref)	1 (33.3)	2 (66.7)	**0.0335**	14.664 (1.234–174.236)
CT+CC	93 (87.7)	13 (12.3)		

Statistically significant differences are in bold; * *p*-values were calculated using the Chi square/Fisher exact test. Stepwise variable selection criterion. N, absolute number of patients; OR, odds ratio; 95% IC, 95% confidence interval for the OR; ALP, alkaline phosphatase; *ABCC2*, ATP Binding Cassette Subfamily C Member 2.

**Table 5 genes-16-00453-t005:** Multivariate logistic regression analysis of *ABCB1* rs2032582 (c.2677A>C/T) with overall survival in non-small lung cancer patients treated with carboplatin and paclitaxel.

Variable	*p*-Value ***	HR (95% CI)
Overall survival (*n* = 91)		
*ABCB1* rs2032582 (c.2677A>C/T)		
CC (ref)		
Non-CC (TT+AA+TA+CA+CT)	**0.0232**	1.922 (1.093–3.377)

Variables (genetic variant, age, sex, ethnicity, smoking, alcoholism, and comorbidities) that, together, increase the risk of death. Statistically significant differences are in bold. * *p*-values were calculated using the Chi square/Fisher exact test. Stepwise variable selection criterion. *n*, absolute number of patients; HR, hazard ratio; 95% IC, 95% confidence interval for the OR; *ABCB1*, ATP Binding Cassette Subfamily B Member 1.

## Data Availability

The original contributions presented in this study are included in the article/Appendix A. Further inquiries can be directed to the corresponding author.

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
