# Peer review of "Association of ABC Efflux Transporter Genetic Variants and Adverse Drug Reactions and Survival in Patients with Non-Small Lung Cancer"

_genes, 2025, doi:10.3390/genes16040453_

Round 1

Reviewer 1 Report

Comments and Suggestions for Authors

The manuscript by Seguin at al. present investigation of the impact of ABCB1 and ABCC2 genetic variants on adverse drug reactions (ADRs) and survival in non-small cell lung cancer (NSCLC) patients treated with carboplatin and paclitaxel.

This study highlights the role of ABCB1 and ABCC2 polymorphisms in chemotherapy response, showing that genetic variations influence drug accumulation, toxicity, and survival outcomes. Pharmacogenomic screening could help tailor chemotherapy regimens, reducing toxicity and improving treatment efficacy in NSCLC patients.

Authors emphasize that identifying high-risk genetic profiles can optimize treatment strategies, reducing ADRs and improving patient quality of life and survival.

The question to be answered/discussed

Why do the rs1128503 CT and CC genotypes, where the CC genotype is associated with normal or increased P-gp activity that promotes drug efflux from cells, show lower overall survival, while the TT genotype may reduce P-gp expression, leading to lower drug efflux and higher intracellular drug accumulation and toxicity, do not show this overall survival?

The limitation of this study is that the data on ADR and OS association are presented for SNPs located in a gene affecting drug pharmacokinetics (drug absorption), while other data on enzymes such as the CYP family responsible for the pharmacodynamics of xenobiotic detoxification are not presented. The expression level of CYP enzymes should be investigated as it may clarify the real situation regarding the SNP association study.

Minor remark

Regarding the number of patients:

initially there were 177 patients. After excluding 65 patients, the total number became 113. Is this correct?

This change will affect the exact calculation in the tables.

Reviewer 2 Report

Comments and Suggestions for Authors

According to the editor’s strict regulation, I have carefully read and checked the article described by Souto Seguin et al. based on its scientific significance, soundness and novelty. In the current study, the authors asked how the genetic variants of ABC1 and ABC2 could cause the development of adverse drug reactions in patients with NSCLC treated with carboplatin and paclitaxel. Although the present study might have certain impact on the related field, there are several concerns (see below) which should be adequately addressed.

Major concerns

The introductive description on the possible involvement of ABCB1/ABCC2 in the development and/or progression of NSCLC was not enough.

The most serious weak point of the present study was a lack of the experimental results. For example, the authors did not show the functional difference between wild-type ABCB1 and ABCB1 variant.

The authors have to clearly highlight the novelty of the current study.

Minor concerns

Total of 110 patients were enough for the present study?

Regarding the possible association between the hematological toxicity and ABCB1/ABCC2 variants, why the current results were completely inconsistent with the previous ones?

ABCB1 variants could dysregulate normal function of P-glycoprotein?

The authors speculated that ABCB1 variant increases the chance of vomiting with carboplatin and paclitaxel treatment. How this ABCB1 variant could induce the vomiting?

Discussion part was too long. The authors have to narrow down the points of discussion, and shorten the length of the discussion section.

For the convenience of the specialized and non-specialized readers, English writing should be further improved.

Comments on the Quality of English Language

For the convenience of the specialized and non-specialized readers, English writing should be further improved.

Round 2

Reviewer 2 Report

Comments and Suggestions for Authors

Introduction and Discussion sections were too long, which makes the readers stressful. The authors have to narrow down the topics of both sections, and shorten their length.
